# Human-Adversarial Visual Question Answering

**Sasha Sheng**[‡][*]   **Amanpreet Singh**[‡][*]   **Vedanuj Goswami**[‡]   **Jose Alberto Lopez Magana**[†]

**Tristan Thrush**[‡]   **Wojciech Galuba**[‡]   **Devi Parikh**[‡][§]   **Douwe Kiela**[‡]
[‡] Facebook AI Research   [†] Tecnológico de Monterrey   [§] Georgia Tech

https://adversarialvqa.org

## Abstract

Performance on the most commonly used Visual Question Answering dataset (VQA v2) is starting to approach human accuracy. However, in interacting with state-of-the-art VQA models, it is clear that the problem is far from being solved. In order to stress test VQA models, we benchmark them against human-adversarial examples. Human subjects interact with a state-of-the-art VQA model, and for each image in the dataset, attempt to find a question where the model's predicted answer is incorrect. We find that a wide range of state-of-the-art models perform poorly when evaluated on these examples. We conduct an extensive analysis of the collected adversarial examples and provide guidance on future research directions. We hope that this Adversarial VQA (AdVQA) benchmark can help drive progress in the field and advance the state of the art.

## 1   Introduction

Visual question answering (VQA) is widely recognized as an important evaluation task for vision and language research. Besides direct applications such as helping the visually impaired or multimodal content understanding on the web, it offers a mechanism for probing machine understanding of images via natural language queries. Making progress on VQA requires bringing together different subfields in AI – combining advances from natural language processing (NLP) and computer vision together with those in multimodal fusion – making it an exciting task in AI research.

Over the years, the performance of VQA models has started to plateau on the popular VQA v2 dataset [20] – approaching inter-human agreement – as evidenced by Fig. 1. This raises important questions for the field: To what extent have we solved the problem? If we haven't, what are we still missing? How good are we really?

An intriguing method for investigating these questions is dynamic data collection [29], where human annotators and state-of-the-art models are put "in the loop" together to collect data adversarially. Annotators are tasked with and rewarded for finding model-fooling examples, which are then verified by other humans. The easier it is to find such examples, the worse the model's performance can be said to be. The collected data can be used to "stress test" current VQA models and serve as the next iteration of the VQA benchmark helping drive further progress.

The commonly used VQA dataset [20] was collected by instructing annotators to "ask a question about this scene that [a] smart robot probably can not answer" [4]. One way of thinking about our proposed human-adversarial data collection is that it explicitly ensures that the questions can not be answered by today's "smartest" models.

---

[*]Equal contribution. Correspondence to advqa@fb.com.
[†]Work done as an intern at Facebook AI Research.

35th Conference on Neural Information Processing Systems (NeurIPS 2021).

Table 1: **Contrastive examples from VQA and AdVQA**. Predictions are given for the VisualBERT, ViLBERT and UniT models, respectively. Models can answer VQA questions accurately, but consistently fail on AdVQA questions.

| Image | VQA | AdVQA |
|---|---|---|
|  | **Q**: How many cats are in the image? 
 **A**: 2 
 **Model**: 2, 2, 2 | **Q**: What brand is the tv? 
 **A**: lg 
 **Model**: sony, samsung, samsung |
|  | **Q**: Does the cat look happy? 
 **A**: no 
 **Model**: no, no, no | **Q**: How many cartoon drawings are present on the cat's tie? 
 **A**: 4 
 **Model**: 1, 1, 2 |
|  | **Q**: What kind of floor is the man sitting on? 
 **A**: wood 
 **Model**: wood, wood, wood | **Q**: Did someone else take this picture? 
 **A**: no 
 **Model**: yes, yes, yes |

This work is, to the best of our knowledge, the first to apply this human-adversarial approach to an image and language multimodal problem. We introduce Adversarial VQA (AdVQA), a large evaluation dataset of 46,807 examples in total, all of which fooled the VQA 2020 challenge winner, the MoViE+MCAN [45] model.

We evaluate a wide range of existing VQA models on AdVQA and find that their performance is significantly lower than on the commonly used VQA v2 dataset [20] (see Table 1). Furthermore, we conduct an extensive analysis of AdVQA characteristics, and contrast with the VQA v2 dataset.

We hope that this new benchmark can help advance the state of the art by shedding important light on the current model shortcomings. Our findings suggest that there is still considerable room for continued improvement, with much more work remaining to be done.

## 2 Related Work

**Stress testing VQA.** Several attempts exist for stress testing VQA models. Some examine to what extent VQA models' predictions are grounded in the image content, as opposed to them relying primarily on language biases learned from the training dataset. The widely used VQA v2 dataset [20] was one attempt at this. The dataset contains pairs of similar images that have different answers to the same question, rendering a language-based prior inadequate. VQA under changing priors (VQA-CP) [2] is a more stringent test where the linguistic prior not only is weak in the test set, but is adversarial relative to the training dataset (*i.e.*, answers that are popular for a question type during training are rarer in the test set and vice versa). Other complementary datasets test model robustness against certain conditions, such as different logicial compositions [18], question

rephrasings [52], suprious correlations [1], and consistency in answering sub-questions [51]. Datasets which benchmark specific capabilities in VQA models also exist, such as reading and reasoning about text in images [57], leveraging external knowledge [44], and spatial [26] or compositional reasoning [28]. Other vision and language datasets, such as Hateful Memes [31], have also tried to make sure the task involves true multimodal reasoning, as opposed to any of the individual modalities sufficing for arriving at the correct label. Compared to the previous work, with AdVQA, we aim to have a more holistic and general dataset that pushes the boundaries of the state-of-the-art on VQA.

**Saturating prior work.** The VQA v2 [20] challenge has been running yearly since 2016 and has seen tremendous progress, as can be seen in Figure 1. From simple LSTM [23] and VGGNet [54] fused models to more advanced fusion techniques (MCB [16], Pythia [27]) to better object detectors (MCAN [61], MoViE+MCAN [45], BUTD [3]) to transformers [59, 56] and self-supervised pretraining (MCAN [61], UNIMO [38]), the community has brought the performance of models on the VQA v2 dataset close to human accuracy, thus starting to saturate progress on dataset. As we show through AdVQA, however, saturation is far from achieved on the overall task, and hence there is a need for new datasets to continue benchmarking progress in vision and language reasoning.

**Adversarial datasets.** As AI models are starting to work well enough in some narrow contexts for real world deployment, there are increasing concerns about the robustness of these models. This has led to fertile research both in designing adversarial examples to "attack" models (e.g., minor imperceptible noise added to image pixels that significantly changes the model's prediction [19]), as well as in approaches to make models more robust to "defend" against these attacks [9, 12]. The human-in-the-loop adversarial setting that we consider in this paper is qualitatively different from the statistical perturbations typically explored when creating adversarial examples. This type of human-in-the-loop adversarial data collection has been explored in NLP e.g., natural language inference [46], question answering [5, 50], sentiment analysis [49], hate

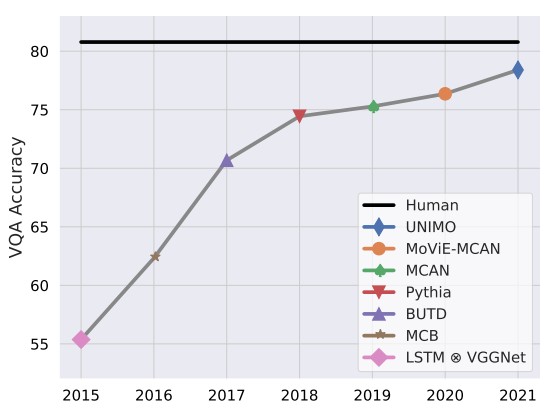

Figure 1: Progress on the VQA v2 dataset over time.

speech detection [60], next video-and-language event prediction [35] and dialogue safety [15]. To the best of our knowledge, ours is the first work to explore an image and language multimodal human-adversarial benchmark.[1]

## 3  AdVQA

The aim of this work is to investigate state of the art VQA model performance via human-adversarial data collection. In this human-and-model-in-the-loop paradigm, human annotators are tasked with finding examples that fool the model. In this case, annotators are shown an image and are tasked with asking difficult but valid questions that the model answers incorrectly. We collected our dataset using Dynabench [29], a platform for dynamic adversarial data collection and benchmarking. For details on our labeling user interface, please refer to the supplementary material.

The VQA v2 dataset is based on COCO [39] images. We collected adversarial questions on both val2017 COCO images and testdev2015 COCO images. We then random sampled the collected set down to 2 questions per val2017 COCO images (10,000) and 1 question per testdev2015 COCO images (36,807). The random sampling was done to balance the annotator's contributions. The data collection involves three phrases (Figure 2): First, in the **question collection** phase, Amazon Mechanical Turk workers interact with a state-of-the-art VQA model and self-report image-question pairs that the model failed to answer correctly. Second, in the **question validation** phase, a new set

---

[1]Concurrent work [36] jointly available with AdVQA at https://adversarialvqa.org also focuses on the same problem but uses images from a different source.

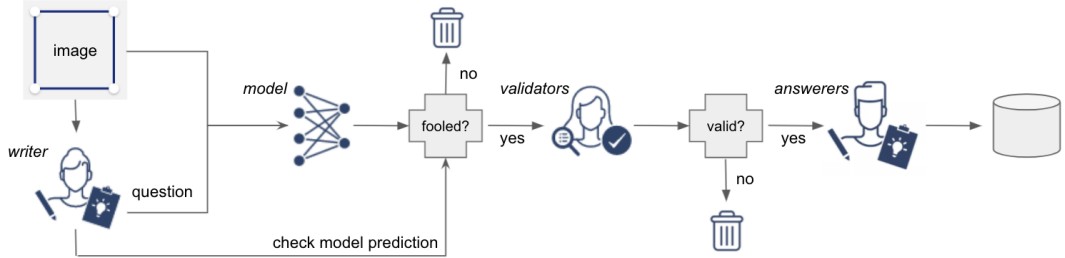

Figure 2: Diagram of the AdVQA human-adversarial data collection flow.

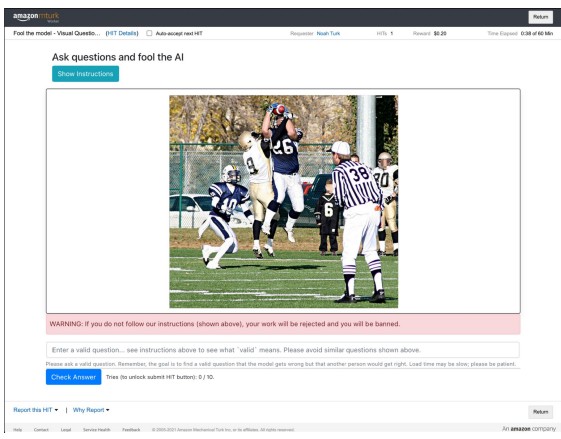

(a) **Main interface**. The annotators follow the instructions provided to write the question in the text field provided which is then answered by the model-in-the-loop.

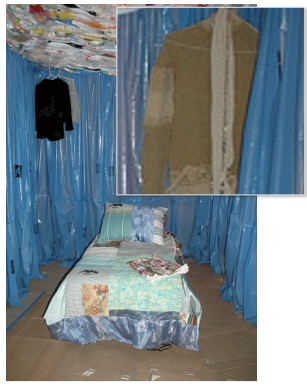

(b) **Magnifier,** available both in question collection, validation and answer collection helps annotators ask and answer more specific questions.

Figure 3: **Question collection interface** showing the first stage of AdVQA collection setup.

of workers validates whether the answers provided by the VQA model for the image-question pairs collected in the first phrase are indeed incorrect. Finally, in the **answer collection** phase, we collect 10 ground truth answers for the image-question pairs validated in the second phase. In what follows, we provide further details for each of these steps.

## 3.1 Question Collection

In this phase, annotators on Amazon MTurk are shown an image and are tasked with asking questions about this image. The interface has a state-of-the-art VQA model in the loop. For each question the annotator asks, the model produces an answer. Annotators are asked to come up with questions that fool the model. Annotators are done (*i.e.*, can submit the HIT) once they successfully fooled the model or after they tried a minimum of 10 questions (whichever occurs first). To account for the fact that it may be hard to think of many questions for some images, annotators are given the option to skip an image after providing non-fooling three questions. We use the VQA Challenge 2020 winner – MoViE+MCAN [45] – trained on the COCO 2017 train set as the model in the loop. This is to ensure that the adversarial benchmark we collect is challenging for the current state-of-the-art in VQA.

The interface provides a digital magnifier to zoom into an area of the image when the mouse hovers over it, allowing workers to examine the image closely if necessary. See Figure 3b. Keyboard shortcuts were provided for convenience.

## 3.2 Question Validation

Note that in the question collection phase, annotators self-report when they have identified a question that is incorrectly answered by the model. Whether or not this was actually the case is verified in the question validation phase. Two different annotators are shown the image, question, and answer

predicted by the model, and asked whether the model's answer is correct. As an additional quality control, the annotator is also asked whether the question is "valid". A valid question is one where the image is necessary and sufficient to answer the question. Examples of invalid questions are: "What is the capital of USA?" (does not need an image) or "What is the person doing?" when the image does not contain any people or where there are multiple people doing multiple things. If the two validators disagree, a third annotator is used to break the tie. Examples are added to the dataset only if at least two annotators agree that the question is valid.

## 3.3 Answer Collection

In the final stage of data collection, following [20, 57], we collect 10 answers per question providing instructions similar to those used in [20] making sure that no annotator sees the same question twice. In addition, as an extra step of caution, to filter bad questions that might have passed through last two stages and to account for ambiguity that can be present in questions, we allow annotators to select *"unanswerable"* as an answer. Further, to ensure superior data quality, we occasionally provide annotators with hand-crafted questions for which we know the non-ambiguous single true answer, as a means to identify and filter out annotators providing poor quality responses.

## 3.4 Human-Adversarial Annotation Statistics

The statistics for the first two stages (question collection and validation) are shown in Table 2. We find that annotators took about 5 tries and on average around 4 minutes to find a model-fooling example. For computing the model error rate, we can look at the instances where annotators claimed they had fooled the model, and where annotators were verified by other annotators to have fooled the model. The latter has been argued to be a particularly good metric for measuring model performance [29]. We also further confirm that the model was indeed fooled by running the model-in-the-loop on a subset of examples in which human agreement was 100% and found the accuracy to be 0.15% on val, and 0.18% on test splits of AdVQA respectively.

Table 2: **AdVQA human-adversarial question collection and dataset statistics.** The model error rate is the percentage of examples where the submitted questions fooled the model (either as claimed during question collection, or after validation). We also report the number of attempts (tries) needed before a validated model-fooling example was found, and how long this took, in seconds.

| Total | Model error rate | | Tries | Time in sec |
|---|---|---|---|---|
| | claimed | validated | mean/median per ex. | |
| 208,932 | 40.94% (85,537) | 36.17% (75,571) | 5.33/4.0 | 203.22/107.26 |

Interestingly, the "claimed" model error rate, based on self-reported model-fooling questions, is similar to that of text-based adversarial question answering, which was at 44.0% for RoBERTa and 47.1% for BERT [5]. The validated error rate for our task is much higher than e.g. for ANLI [46], which was 9.52% overall, suggesting that fooling models is a lot easier for VQA than for NLI. It appears to be not too difficult for annotators to find examples that the model fails to predict correctly.

# 4 Model Evaluation

## 4.1 Baselines and Methods

We analyze the performance of several baselines and a wide variety of state-of-the-art VQA models on the AdVQA dataset. We evaluate the same set of models on VQA v2 dataset as a direct comparison.

**Prior baselines.** We start by evaluating two prior baselines: answering based on the overall majority answer, or the per answer type majority answer in the validation dataset. We use the same answer types as in [4]. The overall majority answer in AdVQA is *no*. The majority answer is *no* for "yes/no", *2* for "numbers" and *unanswerable* for "others". See Section 5 for more details on answer types.

**Unimodal baselines.** Next, we evaluate two unimodal pretrained models: i) ResNet-152 [22] pretrained on Imagenet [13]; and ii) BERT [14], both finetuned on the task using the visual (image)

or textual (question) modality respectively, while ignoring the other modality. We observe that the unimodal text model performs better than unimodal image for both VQA and AdVQA.

**Multimodal methods.** We evaluate two varieties of multimodal models: i) unimodal pretrained and ii) multimodal pretrained. In the unimodal pretrained category we explore MMBT [30], MoViE+MCAN [45] and UniT [24]. These models are initialized from unimodal pretrained weights: BERT pretraining for MMBT; Imagenet + Visual Genome [34] detection pretraining for MoViE+MCAN; and Imagenet + COCO [40] detection pretraining for the image encoder part and BERT pretraining for the text encoder part in UniT. In the multimodal pretrained category, we explore VisualBERT [37], VilBERT [42, 43], ViLT [32], UNITER[11] and VILLA [17]. These models are first initialized from pretrained unimodal models and then pretrained on different multimodal datasets on proxy self-supervised/semi-supervised tasks before finetuning on VQA. VisualBERT is pretrained on COCO Captions [10]; VilBERT is pretrained on Conceptual Captions [53]; ViLT, UNITER and VILLA models are pretrained on COCO Captions [10] + Visual Genome [34] + Conceptual Captions [53] + SBU Captions [47] datasets.

We find that multimodal models in general perform better than unimodal models, as we would expect given that both modalities are important for the VQA task.

**Multimodal OCR methods.** As we will see in Section 5, a significant amount of questions in AdVQA can be answered using scene text.

We test a state-of-the-art TextVQA [55] model, M4C [25] on AdVQA. We evaluate two versions: (i) trained on VQA 2.0 dataset, and (ii) trained on TextVQA [55] and STVQA [6]. In both cases, we use OCR tokens extracted using the Rosetta OCR system [7]. We also use the same answer vocabulary used by other models for fair comparison.

Table 3: **Model performance on VQA v2 and AdVQA** val and test sets. * indicates that this model architecture (but not this model instance) was used in the data collection loop.

| Model | | VQA test-dev | AdVQA test | VQA val | AdVQA val |
|---|---|---|---|---|---|
| *Human performance* | | 80.78 | 89.01 | 84.73 | 88.46 |
| *Majority answer (overall)* | | - | 16.79 | 24.67 | 16.98 |
| *Majority answer (per answer type)* | | - | 31.86 | 31.01 | 33.38 |
| Model in loop | MoViE+MCAN [45] | 73.58 | 13.89 | 73.51 | 14.08 |
| Unimodal | ResNet-152 [22] | 26.66 | 20.59 | 24.85 | 19.02 |
| | BERT [14] | 43.59 | 30.24 | 43.71 | 31.89 |
| Multimodal (unimodal pretrain) | MoViE+MCAN* [45] | 69.81 | 30.02 | 69.77 | 31.31 |
| | MMBT [30] | 49.27 | 30.80 | 49.36 | 32.57 |
| | UniT [24] | 64.34 | 32.12 | 64.32 | 33.94 |
| Multimodal (multimodal pretrain) | VisualBERT [37] | 70.40 | 31.96 | 69.98 | 28.09 |
| | ViLBERT [42] | 59.45 | 32.01 | 59.78 | 33.67 |
| | ViLT [32] | 62.30 | 31.00 | 62.33 | 32.48 |
| | UNITER$_{Base}$ [11] | 70.67 | 27.56 | 69.30 | 29.44 |
| | UNITER$_{Large}$ [11] | 73.58 | 29.66 | 72.82 | 32.08 |
| | VILLA$_{Base}$ [17] | 71.17 | 27.55 | 69.87 | 29.36 |
| | VILLA$_{Large}$ [17] | 72.02 | 28.59 | 71.1 | 30.58 |
| Multimodal (unimodal pretrain + OCR) | M4C (TextVQA+STVQA) [25] | 32.89 | 33.84 | 31.44 | 34.05 |
| | M4C (VQA v2 train set) [25] | 67.66 | 36.57 | 66.21 | 36.93 |

## 4.2 Discussion

Our model evaluation results show some surprising findings. We discuss our observations and hypotheses around these findings in this section.

**Baseline comparison.** Surprisingly, most multimodal models are unable to outperform a unimodal baseline (BERT [14]) and simple majority answer (per answer type) prior baseline which only predicts the most-frequently occurring word for the question's category (*no* for "yes/no", *2* for "numbers" and

*unanswerable* for "others"). A detailed breakdown of category-wise performance provided in Table 4 suggests that even though "yes/no" questions are often considered to be easy, the baseline multimodal models we evaluate are unable to beat the majority answer prediction. We also observe varied but close to majority answer performance in the "numbers" category, which is a question type known to be difficult for VQA models [45]. In contrast, the models outperform the majority answer in the "others" category even though "unanswerable" (the majority answer) is not in the answer vocabulary used. Interestingly, M4C outperforms all on "numbers" and "others" categories possibly thanks to its text reading capabilities. These trends showcase the difficulty of AdVQA and suggest that we have a long way to go still, given that we are as yet apparently unable to beat such simple baselines.

**Model rankings.** First, M4C [25] performs the best among the evaluated models. Interestingly, it is smaller than many of the more sophisticated model architectures that score higher on VQA. This is probably due to the importance of the ability to read and reason about text in the image for answering some AdVQA questions. Second, among models that can't read text, UniT [24] is the best model, despite (or perhaps because of?) it being only unimodal pretrained. As UniT was trained jointly on unimodal as well as multimodal tasks, perhaps, it is less prone to biases in multimodal datasets. Third, the adversarially-trained VILLA [17] model performs surprisingly poorly. While it may be more robust to statistically-generated adversarial examples, it appears to be less so against human-adversarial examples. Fourth, we find that all of these models perform poorly compared to humans, while model performance on VQA is much closer to that of humans. A more detailed analysis on model capabilities required for AdVQA and BERT's superior performance can be found in Section B.2 and B.3.

**Human performance.** Another surprising finding is that inter-human agreement is higher on the AdVQA dataset than on VQA. This could be due to different data annotation procedures, requirements on annotators or just statistical noise. Human-adversarial questions may also be more specific due to annotators having to make crisp decisions about the model failing or not.

**Model-in-the-loop's performance.** Interestingly, the MoViE+MCAN model that was *not* used in the loop and trained with a different seed, performs very similarly to other models. This suggests that to some extent, annotators overfit to the model instance. An alternative explanation is that model selection for all evaluated models was done on the AdVQA validation set, which was (obviously) not possible for the model in the loop used to construct the dataset. In Adversarial NLI [46], the entire model class

Table 4: **The category-wise performance of VQA models.** The state-of-the-art VQA models perform very close to the majority class prior, illustrating the challenge and difficulty of AdVQA.

| Model | Question Type | | |
|---|---|---|---|
| | yes/no | numbers | others |
| Majority Class | 65.87 | 35.55 | 8.83 |
| ResNet-152 | 65.72 | 0.35 | 0.19 |
| BERT | 67.25 | 30.27 | 14.39 |
| VisualBERT | 56.77 | 37.06 | 21.23 |
| ViLBERT | 57.09 | 35.19 | 20.00 |
| MoViE+MCAN* | 50.42 | 34.53 | 19.26 |
| M4C (VQA2) | 64.66 | 38.01 | 21.62 |

of the in-the-loop model was affected. Note however that all VQA models perform poorly on AdVQA, suggesting that the examples are by and large representative of shortcomings of VQA techniques overall, and not of an individual model instance or class.

**Train vs Test Distribution.** We experiment with finetuning VisualBERT and VilBERT further on the AdVQA *val* set, finding that this improved test accuracy from 31.96 to 39.49 and 32.01 to 38.11, respectively, suggesting a difference in the VQA and AdVQA distributions, as we would expect.

## 4.3 Training Details

We train all the models in Table 3 on the VQA *train + val* split excluding the COCO 2017 validation images. We also add questions from Visual Genome [34] corresponding to the images that overlap with the VQA training set to our training split. The VQA *val* set in our results contains all questions associated with the images in the COCO 2017 validation split. This is also consistent with the training split used for the model in the loop. We collect our AdVQA validation set on the images from the COCO 2017 *val* split, ensuring there is no overlap with the training set images. We choose the best checkpoint for each model by validating on the AdVQA validation set.

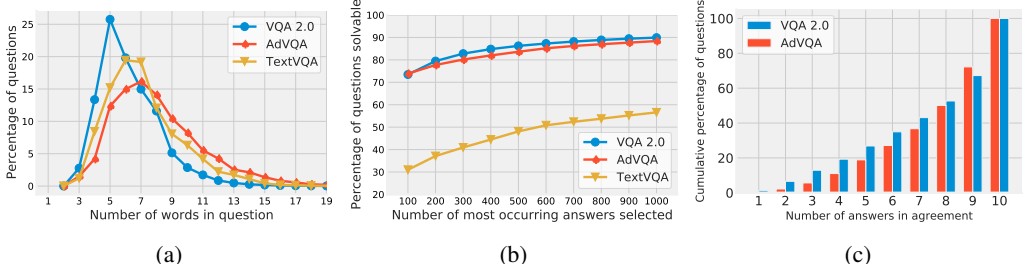

(a)          (b)          (c)

Figure 5: **Quantitative statistics for AdVQA val set questions and answers** showing longer question length, more diversity and better human-agreement. (a) Percentage of questions solvable with a particular question length in AdVQA val set. We see that the average question length (7.82) in AdVQA is higher than the prior work. (b) Percentage of questions solvable when a particular number of top k most occurring answers are selected from each dataset. The plot suggests that AdVQA has a much more diverse answer vocabulary compared to [20] but also not quite as challenging as [57]. (c) Cumulative human agreement scores. We see that human agreement is better and higher on AdVQA compared to [20].

For all our experiments, we use the standard architecture for the models as provided by their authors. For the models that are initialized from pretrained models (whether unimodal or multimodal pretrained), we use off-the-shelf pretrained model weights and then finetune on our training set. We do not do any hyperparamter search for these models and use the best hyperparams as provided by respective authors. We finetune each model with three different seeds and report average accuracy.

We run most of our experiments on NVIDIA V100 GPUs. The maximum number of GPUs used for training is 8 for larger models. Maximum training time is 2 days. More details about hyperparameters, number of devices and time for training each model are provided in the supplementary material.

## 5  Dataset Analysis

**Questions.** We first analyze the question diversity in AdVQA and compare them with popular VQA datasets. AdVQA contains 46,807 questions (10,000 in val and 36,807 in test) each with 10 answers.

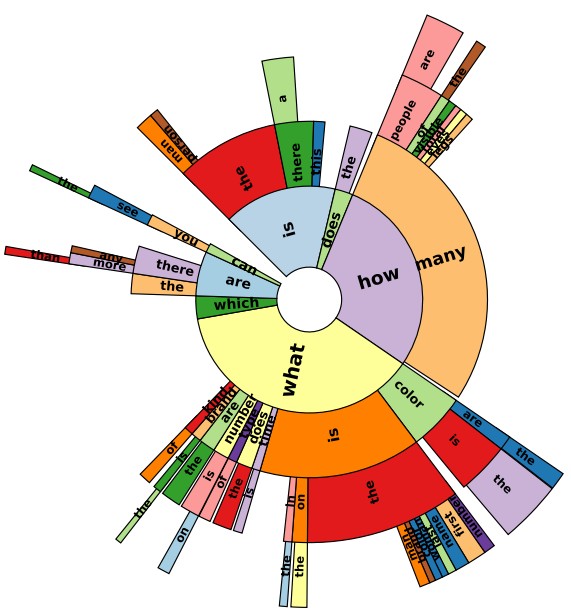

Figure 4: **Sunburst distribution for the first four words in the AdVQA val set questions.** Most questions start with "what" or "how".

Fig. 5a shows the distribution of question length compared with [20, 57]. The average question length in AdVQA is 8.1, which is higher than the VQA v2 (6.3), and TextVQA (7.2). The workers often need to get creative and more specific to fool the model-in-the-loop, leading to somewhat longer questions (*e.g.* specifying a particular person to ask about). Fig. 6a shows the top 15 most occurring questions from the val set, showcasing that questions involving text (*e.g.* time, sign) and counting (*e.g.* how many) are major failures for current state-of-the-art VQA models, corroborating the findings of prior work in [57, 21, 58, 28]. Fig. 4 shows a sunburst plot for the first 4 words in the AdVQA val set questions. We can observe that questions in AdVQA often start with "what" or "how" frequently inquiring about aspects like "many" (count) and "brand" (text).

**Answers.** In AdVQA val set, 75.4% (1,930) answers only occur once compared to 56.2% in VQA v2 [20], suggesting that the diversity of possible answers is much larger in AdVQA compared to

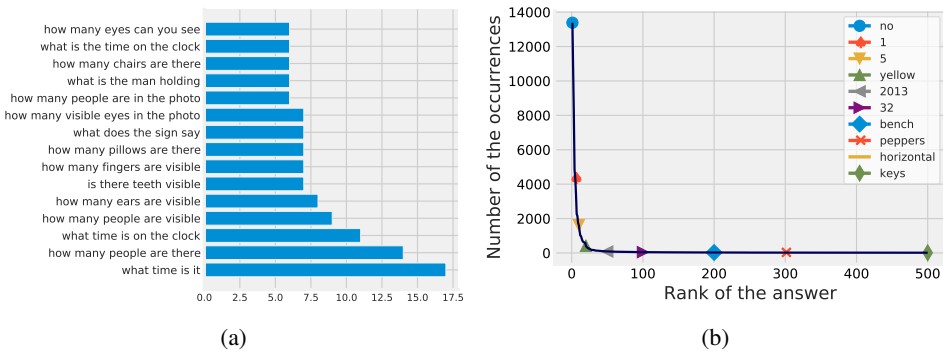

|   |   |
|:-:|:-:|
| (a) | (b) |

Figure 6: **Qualitative statistics for AdVQA val set questions and answers** showing top questions, answers and word distribution. (a) **Top 15 most occurring questions in AdVQA val.** Most of the top questions start with "what". (b) **Total occurrences for 500 most common answers** with markers for particular ranks.

VQA v2. Fig. 5b shows the percentage of questions that are solvable with a vocabulary of top k most occurring answers. We observe that more questions in VQA v2 while fewer question in TextVQA are solvable with smaller vocabulary compare to AdVQA. This suggests that AdVQA is more diverse and difficult than VQA v2 but not as narrowly focused as TextVQA, making it a great testbed for future VQA models. We also showcase more qualitative examples in our supplementary to demonstrate that AdVQA's diversity doesn't lead to unnatural questions. Fig. 5c shows the cumulative percentage of questions where more than a particular number of annotators agree with each other. Fig. 6b shows the top 500 most occurring answers in the AdVQA val set; starting from very common answers such as "no" and counts ("1", "5"), to gradually more specific and targeted answers like "32", "peppers" and "horizontal".

**Answer Vocabulary.** To showcase the challenge of AdVQA, we take the original VQA v2 vocabulary used in the Pythia v0.3 model [55]. We find that 87.2% of the AdVQA val set's questions are answerable using this vocabulary, suggesting that a model with powerful reasoning capability won't be heavily limited by vocabulary on AdVQA. But, we also note that for high performance on AdVQA, a model will need to understand and reason about rare concepts, as 52.6% of the answers in AdVQA val and test sets don't occur in VQA v2 train set.

**Question Types.** Table 5 shows the category-wise distribution for AdVQA questions compared with VQA v2 [20]. We can observe a shift

Table 5: **The category-wise distribution of answers.** Compared to VQA, AdVQA contains more "number"based and lesser "yes/no" questions supporting the prior work's observations around failure of VQA models to count and read text.

| Question Type | VQA test-dev | AdVQA test | VQA | AdVQA val |
|---|---|---|---|---|
| yes/no | 38.36 | 23.22 | 37.70 | 24.58 |
| number | 12.31 | 35.73 | 11.48 | 32.44 |
| others | 49.33 | 41.05 | 50.82 | 42.98 |

from more easy questions of the "yes/no" category in VQA v2 dataset to more difficult questions in "numbers" category (as suggested in prior work [45]) in AdVQA. Please refer to Section B.1 for further detailed breakdown of question types.

**Human Agreement.** In the AdVQA val set, 27.8% human annotators agree on all 10 answers while 3 or more annotators agree an answer for 97.8%, which is higher compared to 93.4% on VQA v2 [20], even though as discussed AdVQA contains a large number of rare concepts.

**Relationship to TextVQA [55].** To understand if the ability to read text is crucial for AdVQA, we extract Rosetta [7] tokens on the AdVQA val set and determine how many questions can be answered using OCR tokens at an edit distance of 1 to account for OCR errors. We find that 11.1% of the AdVQA val questions are solvable using OCR tokens suggesting that ability to read scene text would play a crucial role for AdVQA. Refer to Section B.4 for comparison with other important datasets.

# 6 Conclusion, Limitations & Outlook

We introduced the AdVQA dataset, a novel human-adversarial multimodal dataset designed for accelerating progress on Visual Question Answering (VQA). Current VQA datasets have started plateauing and are approaching saturation with respect to human performance. In this work, we demonstrate that the problem is far from solved.

In particular, our analysis and model evaluation results suggest that current state-of-the-art models underperform on AdVQA due to a reasoning gap incurred from a combination of (i) inability to read text; (ii) inability to count; (iii) heavy bias towards the VQA v2 question and answer distribution; (iv) external knowledge; (v) rare unseen concepts; and (vi) weak multimodal understanding. We've shown the gap is unlikely to be due to (i) limited answer vocabulary; (ii) language representation (BERT performance compared to other); (iii) no pretraining (UniT); or (iv) lack of adversarial training (VILLA performance). The evaluation benchmark for AdVQA is available at https://adversarialvqa.org for the community and we hope that AdVQA will help bridge the gap by serving as a dynamic new benchmark for visual reasoning with a large amount of headroom for further progress in the field.[2]

In future work, it would be interesting to continue AdVQA as a dynamic benchmark. If a new state of the art emerges, those models can be put in the loop to examine how we can improve even further.

## Broader Impact

This work analyzed state-of-the-art Visual Question Answering (VQA) models via a dataset constructed using a dynamic human-adversarial approach. We hope that this work can help make VQA models more robust.

VQA datasets contain biases, both in the distribution of images in these datasets, as well as the corresponding questions and answers which are likely amplified by the VQA models trained on these datasets. Biases studied in the context of image captioning [8, 62] are also relevant for VQA. English is the only language represented in this work and most annotators are based in the United States.

VQA models can be useful for aiding visually impaired users. The commonly used VQA datasets are not representative of the needs of visually impaired users – both in terms of the distribution of images (typically consumer photographs from the web), and in terms of the questions contained in the datasets (typically asked by sighted individuals while looking at the image). In contrast, the VizWiz dataset [21] contains questions asked by visually impaired users on images taken by them to accomplish day-to-day tasks. It is also worth mentioning to the researchers working on the VQA problem that VQA systems can also possibly be used in surveillance systems. It would be beneficial for the community to collect larger datasets in this context to enable progress towards useful and relevant technology while making efforts to reduce the harmful side-effects and applications. It would be important to do this with input from all relevant stakeholders, and in a responsible and privacy-preserving manner. VQA models are currently far from being accurate enough to be uesful or safe in these contexts.

## Acknowledgments and Disclosure of Funding

We thank our collaborators in the Dynabench team for their support.

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
