# Human-Adversarial Visual Question Answering
## (Supplementary Material)

## A   Training Details

Except UNITER [11] and VILLA [17], we trained all models using MMF [55]. The evaluation results with the standard deviation over three different runs with three different seeds are provided in Table A.1. Below we detail the finetuning setup on the COCO [39] train 2017 split evaluated on the AdVQA validation split. Unless otherwise specified, we used the AdamW [33, 41] optimizer with an initial learning rate of $5e-5$, epsilon of $1e-8$, cosine schedule and a warmup of 2000 steps. All jobs are trained in distributed fashion using PyTorch [48]› on NVIDIA V100 GPUs.

Table A.1: **Model performance on VQA v2 and AdVQA** val and test sets. $^*$ indicates that this model architecture (but not this model instance) was used in the data collection loop.

| Model | | VQA test-dev | AdVQA test | VQA val | AdVQA val |
|---|---|---|---|---|---|
| *Human performance* | | 80.78 | 91.18 | 84.73 | 87.53 |
| *Majority answer (overall)* | | - | 13.38 | 24.67 | 11.65 |
| *Majority answer (per answer type)* | | - | 27.39 | 31.01 | 29.24 |
| Model in loop | MoViE+MCAN [45] | 73.56 | 10.33 | 73.51 | 10.24 |
| Unimodal | ResNet-152 [22] | 26.37±0.38 | 10.85±0.37 | 24.82±0.27 | 11.22±0.23 |
| | BERT [14] | 39.47±2.92 | 26.90±0.36 | 39.40±3.23 | 23.81±0.86 |
| Multimodal (unimodal pretrain) | MoViE+MCAN$^*$ [45] | 71.36±0.27 | 26.64±0.45 | 71.31±0.13 | 26.37±0.49 |
| | MMBT [30] | 58.00±4.10 | 26.70±0.24 | 57.32±3.75 | 25.78±0.34 |
| | UniT [24] | 64.34±0.05 | 32.12±0.21 | 64.32±0.04 | 33.94±0.18 |
| Multimodal (multimodal pretrain) | VisualBERT [37] | 70.37±0.05 | 28.70±0.36 | 70.05±0.11 | 28.03±0.33 |
| | ViLBERT [42] | 69.42±0.30 | 27.36±0.18 | 69.27±0.14 | 27.36±0.32 |
| | ViLT [32] | 64.52±0.42 | 27.11±0.14 | 65.43±2.43 | 27.19±0.50 |
| | UNITER$_{Base}$ [11] | 71.87±0.01 | 25.16±0.49 | 70.50±0.08 | 25.20±0.19 |
| | UNITER$_{Large}$ [11] | 73.57±0.21 | 26.94±0.31 | 72.71±0.22 | 28.03±0.33 |
| | VILLA$_{Base}$ [17] | 70.94±1.25 | 25.14±0.93 | 69.50±1.44 | 25.17±0.67 |
| | VILLA$_{Large}$ [17] | 72.29±0.39 | 25.79±0.46 | 71.40±0.37 | 26.18±0.15 |
| Multimodal (unimodal pretrain + OCR) | M4C (TextVQA+STVQA) [25] | 32.89±0.57 | 33.84±0.29 | 31.44±0.59 | 34.05±0.34 |
| | M4C (VQA v2 train set) [25] | 67.66±0.34 | 36.57±0.36 | 66.21±0.38 | 36.93±0.21 |

**MoViE+MCAN** We use a batch size of 64 for 236K updates using a multi-step learning rate scheduler with steps at 180K and 216K, learning rate ratio of 0.2 and a warmup for 54K updates. Training takes an average of 2 days.

**Unimodal** We train the models with a batch size of 64 for 88K updates with linear learning rate schedule starting from $1e-5$ with a warmup for 2000 updates. We used a linear learning rate schedule with 2000 warm up steps. The training takes an average of 8 hours.

**MMBT [30]** We trained MMBT from scratch with a batch size 64 without any pretraining following [30] for 88K updates. The training takes an average of 17 hours.

**UniT [24]** We initialized from the model pretrained on all 8 datasets [24] with COCO initialization. We set the batch size to 8, weight decay as $1e-4$ and train the model on 8 GPUs for 2 days.

**VisualBERT [37]** We trained VisualBERT from the best pretrained model on COCO using MLM loss using a batch size of 64 which takes an average of 8 hours.

**ViLBERT [42]** We trained ViLBERT from the best pretrained model on Conceptual Captions using MLM loss using a batch size of 64 which takes an average of 13 hours.

**ViLT [32]** We trained ViLT with 44K updates, initial learning rate of $1e-4$, eps as $1e-8$ and weight decay as 0.01 for an average of 7 hours.

**UNITER/VILLA [11, 17]** We used the author-provided pretrained checkpoint for UNITER[3] and VILLA[4] with a confidence threshold of 0.075. The rest of the hyper parameters were consistent with the configuration provided in the repository. The training takes about 2 hours.

**M4C [25]** We used the same training schedule and hyper parameters as [25] used for TextVQA training [57] for both TextVQA + STVQA and VQA v2. We didn't use extra Visual Genome [34] QA data due to lack of OCR text and used the OCR data from [25] for both COCO and TextVQA.

## B    Extended Dataset and Results Analysis

### B.1    Question Types

To check what kind of questions types are present in AdVQA, we provide a detailed breakdown of question types based on categories similar to [4] in Table B.1.

Table B.1: **Detailed question type breakdown** for AdVQA's full validation set along with percentage of those questions in "other"

| Question Type | full val set (%) | "others" |
|---|---|---|
| what is | 15.14 | 31.36 |
| what color | 5.57 | 12.96 |
| what kind | 0.85 | 1.98 |
| what are | 1.49 | 3.19 |
| what type | 0.85 | 1.98 |
| is the | 1.29 | 4.98 |
| is this | 1.02 | 0.35 |
| how many | 27.92 | 0.51 |
| are | 6.35 | 1.77 |
| does | 2.57 | 0.30 |
| where | 6.4 | 1.49 |
| is there | 0 | 0 |
| why | 0.03 | 0.07 |
| which | 3.26 | 7.40 |
| do | 0.67 | 0.02 |
| what does | 1.47 | 3.40 |
| what time | 0.76 | 1.77 |
| who | 0.36 | 0.84 |
| what sport | 0.12 | 0.28 |
| what animal | 0.32 | 0.74 |
| what brand | 0.93 | 2.16 |

### B.2    Model Capabilities Required

To understand what capabilities are required for answering questions in AdVQA, we also sampled 50 random examples from the validation set. If more than one category is required for answering the questions, we include the question in all those categories. The categorization is based on the answer (i.e., what answering the question requires), rather than the question. Based on our analysis, we can see that AdVQA does require multiple kinds of reasoning capabilities in a model to solve it:

- 10 questions required color understanding.
- 5 questions' answers were common objects (80 common object classes in COCO)
- 16 questions required reading or understanding text for answering
- 17 questions required positional understanding and reasoning.
- 5 questions required understanding uncommon objects
- 5 questions required external knowledge in some form.

---

[3]UNITER Code: https://github.com/ChenRocks/UNITER
[4]VILLA Code: https://github.com/zhegan27/VILLA

- At least 4 questions were directly solvable by common sense reasoning.
- One question required counting the number of the objects in the image.

## B.3    BERT's Performance

In comparison to multimodal models, BERT is very good at picking up on natural "biases" in AdVQA, which it probably learns from its pretraining on large scale text corpora. As seen in Table 4, the category-wise performance of BERT compared to multimodal is different as multimodal models perform better on numbers and others category while lagging on yes/no category. BERT's performance is similar to majority class performance and majority class performance on all categories suggesting that BERT picks up on the dataset biases very well. We confirm this by doing an extensive analysis on BERT and VisualBERT's predictions on AdVQA:

- 66.6% of the times BERT predicts "no" majority answer compared to 54.1% of the times for VisualBERT when the question is of "yes/no" type.
- For numeric questions, 45.6% times BERT predicts majority answer "2" whereas VisualBERT only predicts it 40.1% times.
- For questions requiring reading text (keywords "written", "sign", "text"), BERT mostly predicts "stop".
- For questions related to brand BERT mostly predicts "nike", "coca-cola", "ford" or "apple".
- For countries, BERT mostly answers "usa".
- For color, mostly predicts "black" or "white".
- For questions asking what the person is holding predicts "tennis racket".
- For cities, BERT mostly predicts "new york".
- For position relation questions, BERT mostly predicts "right".

## B.4    Relationship to OKVQA, TextVQA and VQA-CP

In contrast to other VQA datasets which either target a very specific failure mode for VQA models or reorganize data distribution to reduce impact of bias, in AdVQA we create a more holistic and general dataset to make VQA more challenging. Specifically, compared to (i) OKVQA [44] in which questions specifically require external knowledge, (ii) TextVQA [57] requiring scene text reading and understanding, and (iii) VQA-CP [2] specifically changing answer distribution between train and test splits, we don't ask annotators to specifically target a fixed failure mode or artificially generate an unbiased question distribution but take a more holistic model-in-the-loop approach. In our opinion, asking annotators to fool an existing VQA model covering all of these failure modes and biases (and more) makes AdVQA more suitable to be the next generation of a generic VQA benchmark. Having these alternative benchmarks has helped certainly push progress and put a magnifying lens on very difficult failure modes individually and we consider them complementary to AdVQA.

# C    Annotation details

Before annotators were able to proceed with the main task, they had to pass an on-boarding phase (Section C.1). We restricted the interface to be only accessible on non-mobile devices, to annotators in the US with at least 100 approved hits and with an approval rate higher than 98%. The annotators were paid a bonus if their self-claimed fooling question was verified to be fooling and valid by two other annotators.

## C.1    Qualification Phase

The annotators were asked to go through a two-stage qualification phase. In the first stage, they were shown 11 examples that include both valid and invalid questions. Figure B.1 shows valid and invalid examples in the Example Stage. After scrolling through those 11 examples, the annotators proceeded to the second stage, in which we ask them to complete a quiz. The annotators passed only if they got more than 6 out of 7 correct, after which they qualified for the main task. There are two types of quiz questions: 1) determine if the provided answer is correct for the specific image question pair; and 2) determine if a given question is valid with the image as context. See Figure B.2 for examples of those

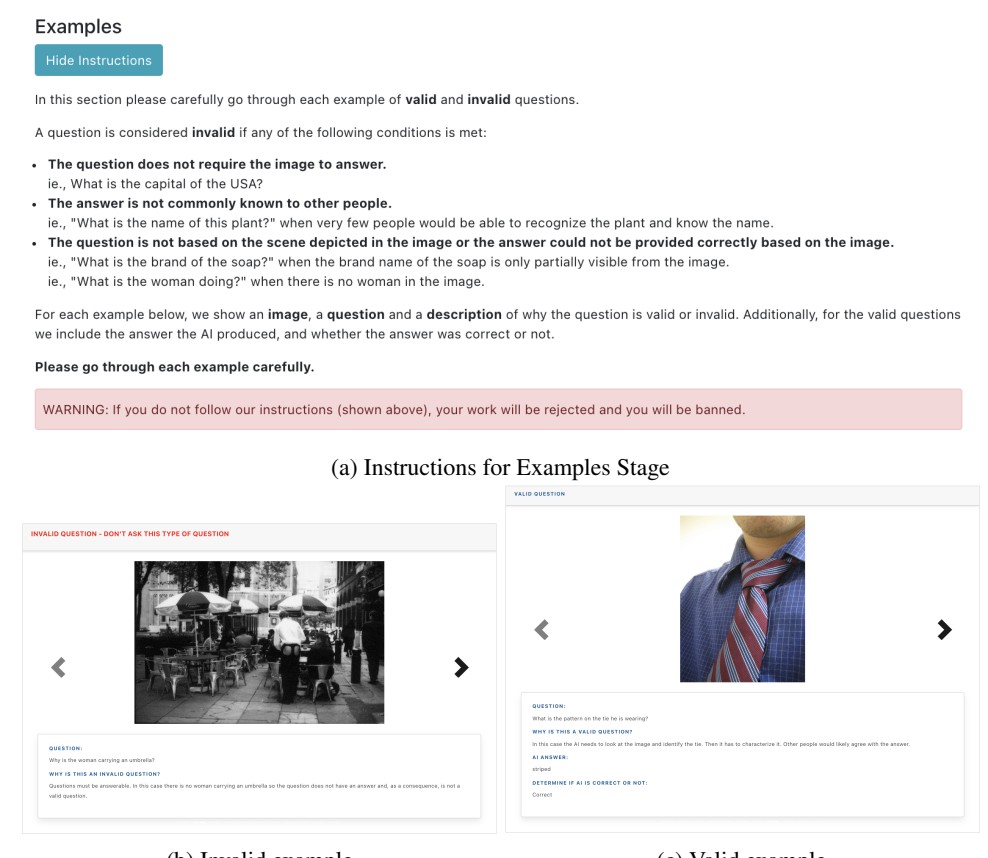

(a) Instructions for Examples Stage

(b) Invalid example

(c) Valid example

Figure B.1: Qualification Phase Stage 1: View Examples

two question types. If an annotator failed the first time, they were given an explanation on the correct choice before being allowed a second try on a different (but similar) set of questions.

## C.2 Main Labeling Task

Figure B.3a and B.3b show the instructions given to first-time annotators for the "question collection" and "question validation" tasks. The instructions were hidden for the non-first-time annotators, but remained accessible via a button at the top. Figure B.4a and B.4b show the preview landing pages on Mechanical Turk.

## C.3 Answer collection task

Figure C.1 and C.2 show the preview and main interface of the answer collection stage. For each question, we collect ten answers from ten different annotators using this interface. We provide explicit instructions following [20] and [57] to avoid ambiguity and collect short relevant answers. The annotators are also provided a checkbox to select "unanswerable" in case the question is ambiguous or can't be answered which annotators are suggested to use sparingly. Finally, we use and show a set of hand-crafted already annotated questions without ambiguity randomly to filter out bad annotators by comparing their answers to ground truth. An annotator is prevented from doing the task if they fail the test three times.

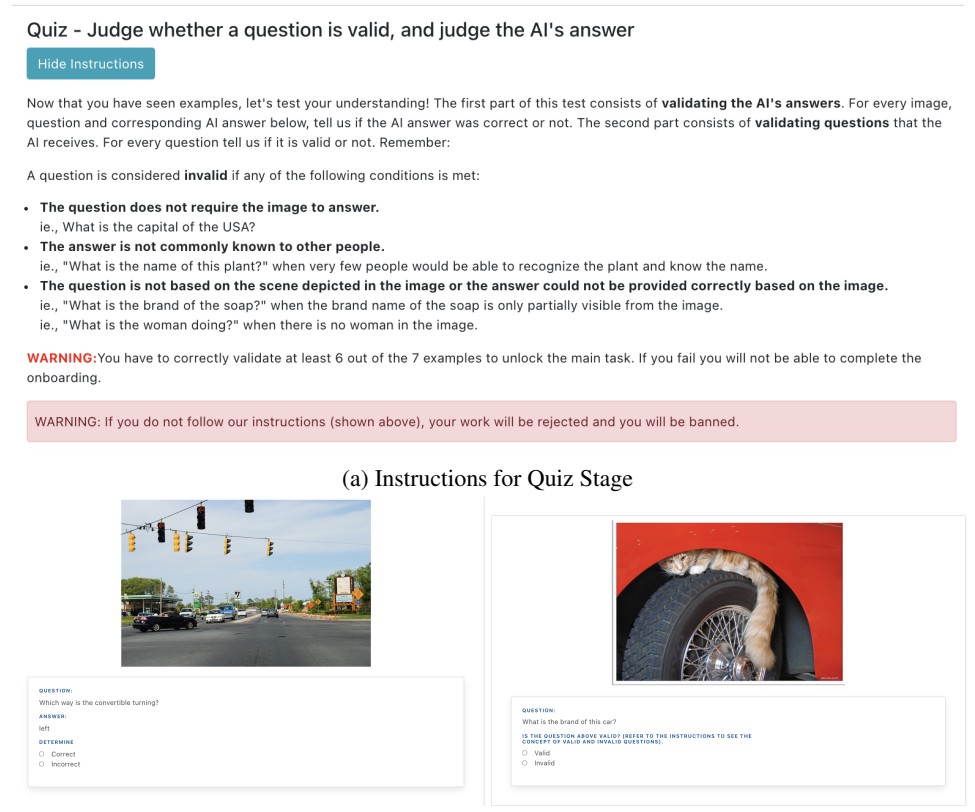

(a) Instructions for Quiz Stage

(b) Determine if the answer is correct

(c) Determine if the question is valid

Figure B.2: Qualification Phase Stage 2 - quiz. Annotators are allowed access to the main task only if they passed the quiz.

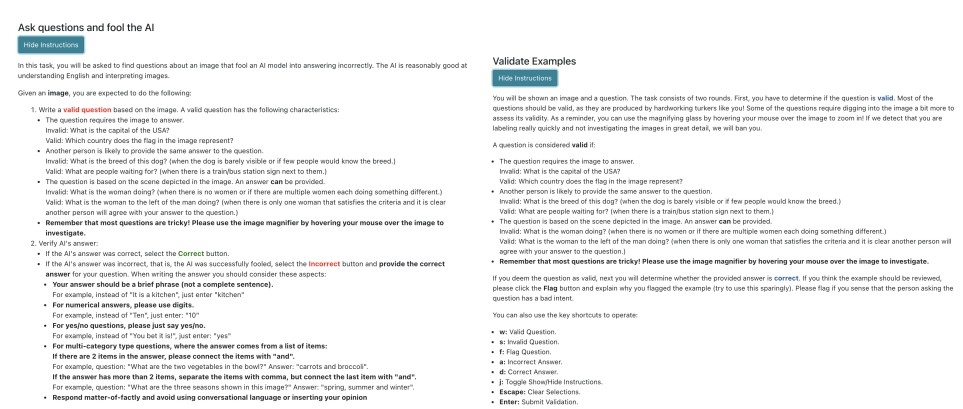

(a) Question Collection Instructions

(b) Question Validation Instructions

Figure B.3: Main Labeling Task Instructions

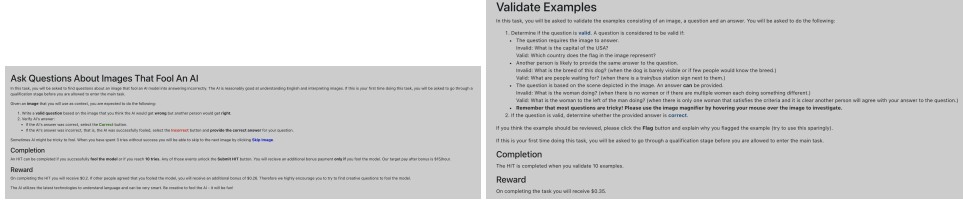

(a) Preview for Question Collection  (b) Preview for Question Validations

Figure B.4: Preview - landing page on MTurk interface.

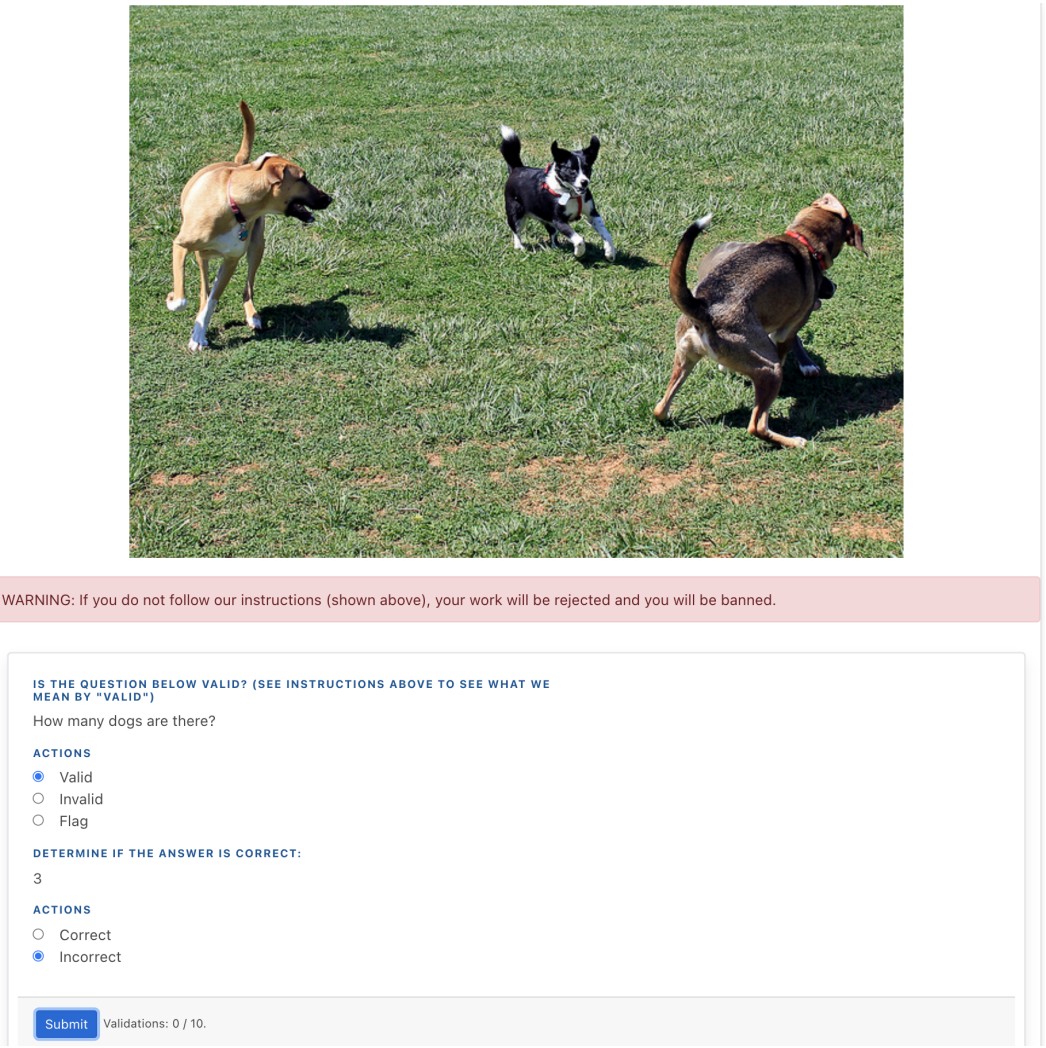

Figure B.5: Validation Interface

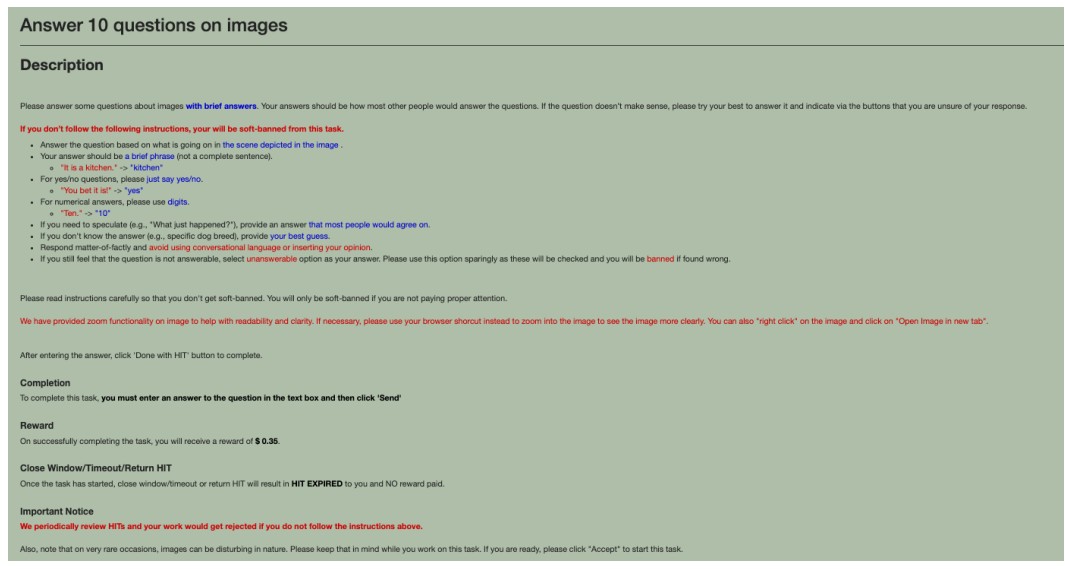

Figure C.1: Preview for Answer Collection

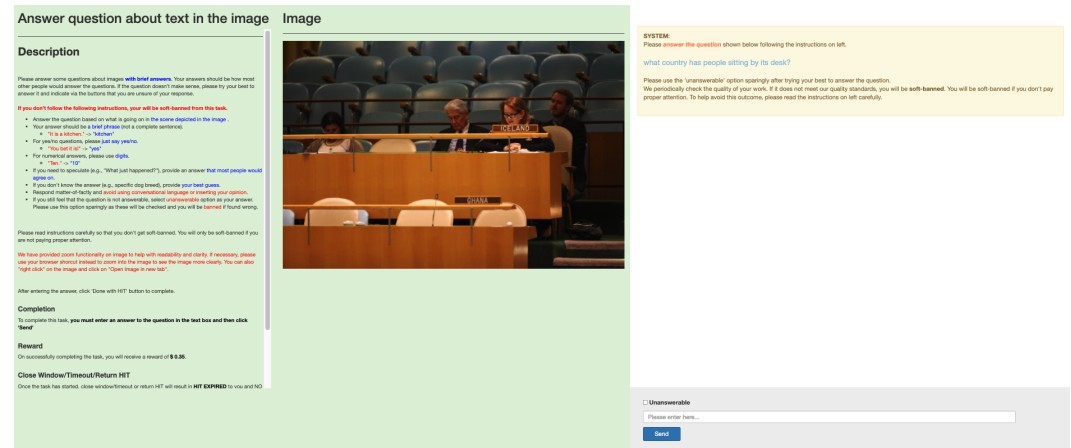

Figure C.2: Answer collection interface

# D   Random Samples

We show 10 randomly selected samples in Figure D.1.

Table D.1: **Random examples from AdVQA**.

| Image | AdVQA |
|---|---|
| 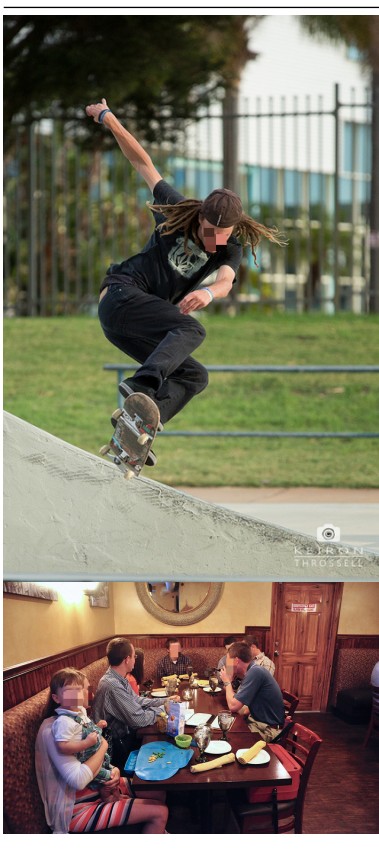 | **Q**: Which hands does he have bracelets on?
**Processed Answers**: both (count: 9)
**Raw Answers**: 'both', 'both hands', 'both', 'both' 'both', 'both', 'both', 'both', 'both', 'both' |
|  | **Q**:What is the baby wearing?
**Processed Answers**: overalls (count: 6)
**Raw Answers**: 'shirt & overall', 'overalls', 'overalls', 'overalls' 'jumper', 'overalls', 'coverals ', 'dungerees', 'overalls', 'overalls' |

| Image | AdVQA |
|-------|-------|

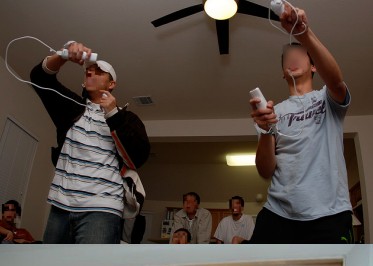

**Q**: How many people can be seen in the room?
**Processed Answers**: 6 (count: 5), 7 (count: 3)
**Raw Answers**: '7', '5', '6', '8', '6', '7', '6', '6', '7', '6'

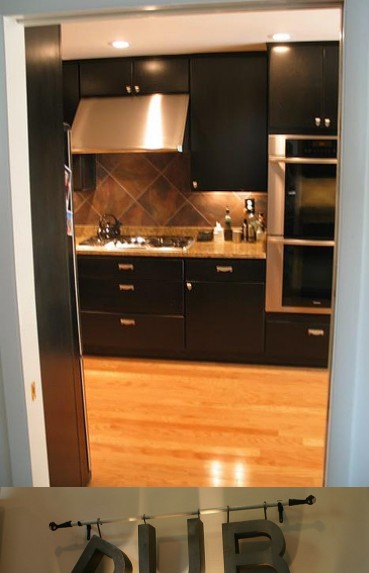

**Q**: What is on the stovetop?
**Processed Answers**: kettle (count: 4)
**Raw Answers**: 'tea kettle', 'kettle', 'teapot', 'teapot', 'right'
'tea kettle', 'kettle', 'teapot', 'kettle', 'kettle'

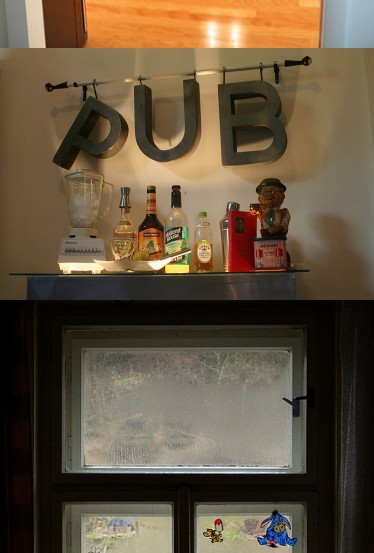

**Q**: Whats does the three letters spell?
**Processed Answers**: unknown (not in vocab)
**Raw Answers**: 'pub', 'pub', 'pub', 'pub', 'pub'
'pub', 'pub', 'pub', 'pub', 'pub'

**Q**: Are the windows all the same size?
**Processed Answers**: no (count: 10)
**Raw Answers**: 'no', 'no', 'no', 'no', 'no', 'no'
'no', 'no', 'no', 'no'

| Image | AdVQA |
|---|---|

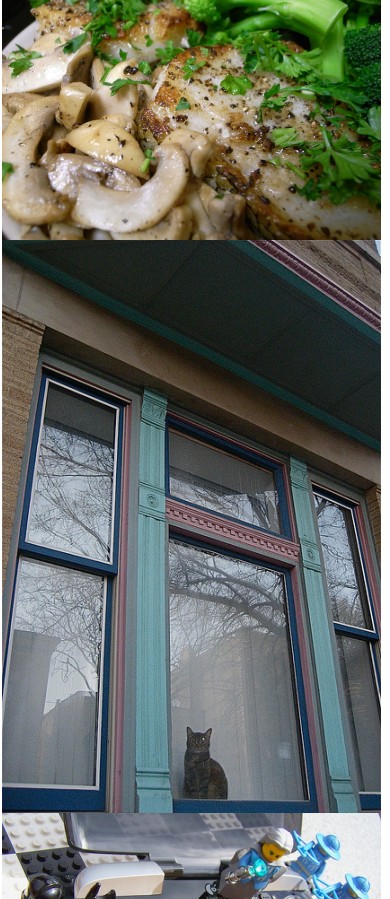

**Q**: how many pieces of meat?
**Processed Answers**: 2 (count: 7), 1 (count: 3)
**Raw Answers**: '2', '2', '2', '2', '2', '2', '1', '1', '1', '2'

**Q**: Which is the largest window?
**Processed Answers**: middle (count: 3)
**Raw Answers**: 'center', 'bottom middle', 'bottom middle',
'middle', 'middle', 'bottom middle', 'middle lower '
'where a cat sits', 'middle one', 'the middle'

**Q**: What color is the checkerboard background?
**Processed Answers**: black and white (count: 5)
**Raw Answers**: 'unanswerable', 'black and white', 'gray'
'black and white', 'black', 'black white', 'black and white'
'black and white', 'black white', 'black and white'

**Q**: What does it say on the side of the boat closest in the foreground?
**Processed Answers**: unknown (not in vocab)
**Raw Answers**: 'sanssouci', 'sanssouci', 'sanssouci'
'sanssouci', 'sanssouci', 'sanssouci', 'sanssouci', 'sanssouci'
'sanssouci', 'sanssouci'