# OpenReview forum: "Human-Adversarial Visual Question Answering"
_NeurIPS.cc/2021/Conference — NeurIPS 2021 Poster_

### Official Review · Reviewer_BJYR · 2021-07-13

**Rating:** 6
**Confidence:** 4

**Summary:**

This paper works on the topic of visual question answering and aims to test how well state-of-the-art models perform on adversarial examples. Specifically, the authors collect a new adversarial benchmark on VQA, named "Adversarial VQA" (AdVQA), including a large evaluation dataset of 28,522 examples in total. The data samples in AdVQA are collected by putting both human annotators and state-of-the-art models in the collection loop: Annotators are requested to create questions that can fool the state-of-the-art models, and these questions are further verified by other annotators. The authors benchmark state-of-the-art models on these collected adversarial questions and their performance drops significantly.

The main contribution of this paper is the collected AdVQA dataset which includes adversarial examples created by human. The experimental results suggest future research directions on VQA.

**Limitations And Societal Impact:**

It is unclear if the collected questions are "out-of-distribution" or "adversarial". The authors might perform some analysis to demonstrate the gap between AdVQA and VQA v2, such as the overlap between word vocabularies of AdVQA and VQA v2, the overlap between answer vocabularies of AdVQA and VQA v2, the answer distribution for each question type (like counting questions), etc.

**Main Review:**

Strengths:
1. It is interesting and novel to collect adversarial examples by making human annotators interact with and fool state-of-the-art models. The questions generated in this way are semantically meaningful to humans, compared with adversarial examples created by adding imperceptible noise.

2. The collected benchmark AdVQA is valuable to the community. It provides a quantitive measure of how well current state-of-the-art models perform on adversarial examples and can further spur the progress along this direction. The authors promise to make a public evaluation server to evaluate on AdVQA.

3. The authors conduct extensive experiments on AdVQA and benchmark tens of baseline methods. The analysis points out several reasons for the performance gap between humans and models, such as inability to read text and count, heavy bias, and external knowledge.

4. The paper is overall well-written and easy to follow.

Weaknesses:
1. My major concern is that the collected examples are more of "out-of-distribution" than "adversarial". As explained in Line 302~303, 15.4% of the questions in AdVQA require detecting texts in images, as shown by the first example in Table 1. The state-of-the-art models that are trained on VQA v2 definitely fail on these questions, since they are never trained on such examples. It is similar to the questions requiring external knowledge or involving unseen concepts.

2. It is unclear what is the most significant difference between AdVQA and other VQA datasets, such as OKVQA (requiring external knowledge), TextVQA (involving texts in images), and VQA-CP (changing prior). Although AdVQA is collected in a human-adversarial manner, the collected questions seems a combination of previous datasets. The authors might clarify if there are some unique points in AdVQA, or if not, why such a combination is meaningful and what benefits it brings compared with previous datasets.


======post rebuttal======

Thank the authors for their responses. After reading these responses and other reviews, I keep my original rating and vote for accept.

**Time Spent Reviewing:**

5

---

> ### Author Response · Authors · 2021-08-10
> **Response to Reviewer BJYR**
>
> Thank you for your thoughtful feedback!
>
> - Regarding whether the examples are more of "out-of-distribution" than "adversarial", we believe this is not the case due to the following two reasons:
>    - Using the answer vocabulary from the Pythia model for the VQA 2.0 dataset, one can achieve 77.2% on the AdVQA validation set. This number is 48.46% for the TextVQA validation set, suggesting that the answer distribution for AdVQA is indeed not “out-of-distribution”, as it is for TextVQA. Furthermore, in Table 3, the best amongst the current state-of-the-art models only gets around 33%, which is far lower than 77.2%, suggesting that there is more at play here than just distributional differences corresponding to lower numbers.
>    - To confirm that AdVQA is indeed not solvable by models which can use scene text, we evaluate M4C [23] on AdVQA in Table 3 and observe a performance gain of only 4%. The gain is actually higher when trained on VQA 2.0 compared to TextVQA, suggesting that AdVQA is more in-domain to VQA compared to TextVQA.
> - Comparison with OKVQA, TextVQA and VQA-CP?
>    - In contrast to other VQA datasets which either target a very specific failure mode for VQA models or reorganize data distribution to reduce impact of bias, in AdVQA we create a more holistic and general dataset to make VQA more challenging. Specifically, compared to (i) OKVQA in which questions specifically require external knowledge, (ii) TextVQA requiring scene text reading and understanding, and (iii) VQA-CP specifically changing answer distribution between train and test splits, we don’t ask annotators to specifically target a fixed failure mode or artificially generate an unbiased question distribution (which can be problematic, as raised by Reviewer bpjd) but take a more holistic model-in-the-loop approach. Asking annotators to fool an existing VQA model covering all of these failure modes and biases (and more) makes AdVQA more suitable to be the next generation of a generic VQA benchmark, in our opinion. That said, having these alternative benchmarks has helped certainly push progress and put a magnifying lens on very difficult failure modes individually - together with AdVQA these datasets can help build better VQA models in the future. We will include a more detailed discussion in the final version, if accepted.

---

### Official Review · Reviewer_MEbN · 2021-07-14

**Rating:** 8
**Confidence:** 4

**Summary:**

This paper proposes AdVQA, a new (adversarial) test set for the Visual Question Answering.  Annotators and a SOTA model are put in the loop to make sure the model can be fooled. Experiments show that other VQA models tend to perform poorly on AdVQA, and a comprehensive analysis discusses the questions and answers that are in this dataset (compared with VQA and TextVQA).

**Limitations And Societal Impact:**

The limitations of this work do not seem to be addressed in the "limitations" section, but rather the Broader Impact section. The main limitation stated is that this dataset isn't as good as others (like VizWiz) for actually aiding the visually impaired, even though this is often mentioned as a goal to this work.

The broader impact statement touches on social biases, but it could flesh this out a bit more (e.g. from what countries are annotators from? what kinds of social biases emerge and from where in the pipeline during your annotation setup?) It could also be improved by discussing end impacts of this work, e.g. downstream use of VQA like models for surveillance applications.

**Main Review:**


To this reviewer, this paper seems strong. It is perhaps not surprising that performance is lower on AdVQA, however, it is impressive that it has higher human agreement than the non-adversarial VQA. The magnitude of how much models drop in performance is significant (at least this reviewer): from the 70% range to 33% at most. It seems like this could be a great addition in the community (assuming it is released, etc.) for modeling work on VQA-V2 to report on AdVQA in addition.

The qualitative analysis is insightful and much appreciated (to this reviewer).

The additional experiments are helpful. Interestingly, the discussion at L222 (suggesting that performance goes up just a little bit if finetuned on val) seems qualitatively different to trends with other datasets. For instance, in a lot of past work (e.g. Adversarial Filtering (from Le Bras et al ICML 2020 and earlier works)) one round of 'fooling the model' is shown to be not enough. However, it's great that it works here (possibly though because val is small).

The limitations and broader impact sections could be somewhat improved (see the next section).

**Time Spent Reviewing:**

1

---

> ### Author Response · Authors · 2021-08-10
> **Response to Reviewer MEbN**
>
> Thank you for your support! We will improve the limitations and broader impact sections as you suggest, specifically by providing more details on the annotators and any social biases that might emerge from the particular setup, and by discussing downstream applications in more detail. Thanks for these suggestions!

---

### Official Review · Reviewer_bpjd · 2021-07-16

**Rating:** 6
**Confidence:** 4

**Summary:**

This paper collects a new validation dataset via crowdsourcing to benchmark the progress of state-of-the-art VQA models.
The dataset (containing 22K questions and 10x answers) is adversarially collected such that annotators have to come up with questions that can successfully fool a state-of-the-art VQA model, and such questions are undoubtedly more challenging to various existing models.
The authors perform thorough data analysis as well as extensive experiments with a variety existing models, to validate the usefulness of the new dataset.
This new dataset can potentially be a valuable benchmark to track the progress of VQA research, if there is no obvious bias in the dataset (potential dataset bias is not thoroughly discussed in this paper).

========================================================================================

Thanks for the response from the authors. I've updated my rating after reading the rebuttal as well as the comments from other reviewers.



**Limitations And Societal Impact:**

* My major concern is about the quality of the collected dataset (i.e., whether it's biased), and the authors also mention this issue in L212-213. Specifically, since only one SOTA model is used as "adversary" for human to fight against, it is very possible that the new dataset can be "biased" towards this specific model checkpoint (i.e. human has the capability to find a "shortcut" to fool a model). For example, annotators may quickly learn that asking questions about embedded text can easily fool the model, thus leading to the new dataset containing many questions regarding text. Also From Table 3, it shows a same architectured but differently seeded model shows similar performance. The authors discussed a bit about this in L219-L221 (by the way it is not surprising that VQA models will perform poorly on AdVQA as simple questions are all ruled out), but more evidence would be needed to better address the concern (quality of the dataset). For example, if a M4C model (capable of reading text) was selected as "adversary model" in the dataset collection phase, would that cause the collected dataset to differ significantly from current version? If an ensemble (containing multiple instances of a variety of different type of models) are used as adversary, probably the collected dataset will be of higher quality. As a benchmark, if the dataset is flawed with bias it may lead the community to an undesired direction. For example, VQA-CP v2 is a benchmark measuring model robustness against priors, but since it's programmatically reversed the answer distribution ("more frequent" in training = "less frequent" in test), some debiasing methods exploit this characteristic and achieves strong performance on VQA-CP v2 benchmark but are not really generalizable [1][2]. Therefore I am leaning towards being conservative and cautious on such benchmarks.

[1] Teney, Damien, et al. "On the value of out-of-distribution testing: An example of goodhart's law." arXiv preprint arXiv:2005.09241 (2020).
[2] Kervadec, Corentin, et al. "Roses Are Red, Violets Are Blue... but Should Vqa Expect Them To?." Proceedings of the IEEE/CVF Conference on Computer Vision and Pattern Recognition. 2021.

**Main Review:**

1. Originality
* In my view, this is mainly a resource paper as the major contribution would be the collected new dataset. The data collection procedure is standard, but collecting high quality dataset is definitely not trivial especially for human-in-the-loop dataset collection.
* The authors claim this is the first work to explore multimodal human-adversarial benchmark.

2. Quality
* Extensive analysis about the new dataset is provided in the paper, and experiments with existing VQA methods show the usefulness of the dataset.
* As a resource paper, technical contribution of this work is limited as it evaluated existing methods off-the-shelf on this new dataset.

3. Clarity
* The paper is well-written and quite easy to follow.

4. Significance
* This benchmark has the potential to be an important "stress test" testbed for new VQA methods, if there is no obvious bias in the collected dataset (which is not thoroughly tested thus unknown from this paper).

**Time Spent Reviewing:**

6

---

> ### Author Response · Authors · 2021-08-10
> **Response to Reviewer bpjd**
>
> Thank you for your thoughtful comments. We absolutely agree that it is important to be cautious, with any kind of benchmark but especially with stress tests, and we appreciate your concerns around potential dataset bias. We agree that the new dataset may be overfitted somewhat to the model that was used in-the-loop during data collection. Our response to this potential issue is twofold:
>
> 1) If the dataset was in fact overfitted to a particular model with highly specific “shortcuts”, the examples would not fool the other models. Instead, we find that all models consistently perform very poorly on the dataset. This suggests that any potential shortcuts are at least present in all of the models we tested, which is a quite large range of models.
>
> 2) One of the beauties of dynamic adversarial data collection is that it explicitly acknowledges that no dataset is perfect: if any bias exists or emerges, annotators and researchers will be able to exploit it eventually, which will lead to better models that can then be put in the loop for further data collection. In this sense, the mechanism we employ can address biases much more cleanly than alternatives like the VQA-CP benchmark you mentioned. That said, to our knowledge, no construction-specific characteristics exist in AdVQA which can be exploited like in VQA-CP.

---

### Official Review · Reviewer_JzmN · 2021-07-17

**Rating:** 6
**Confidence:** 4

**Summary:**

Update after rebuttal:
Thanks to the authors for their rebuttal in response to my questions/concerns. After reading the rebuttal and the other reviews, I am updating my score to 6, to better reflect the contributions made by the work.
_______________________________________________________________________________________________________________
The paper introduces an adversarial VQA dataset collected with human-and-model in the loop by directly asking humans to write questions to attack the winning model of VQA challenge. Extensive evaluation of existing models on this adversarial VQA dataset suggest that current VQA systems are far from really solving the VQA problem. Many well-performed models on VQA v2 seem to fail on AdVQA. AdVQA potentially can be used to benchmark the advances of VQA models in the future, considering VQA v2 performance is close to human parity.

**Limitations And Societal Impact:**

Yes. It is included in the last section and Broader Impact section.

**Main Review:**

`Missing discussions on related VQA stress test datasets`

There are many other VQA datasets built upon VQA v2 to stress test VQA models. For example, VQA-LOL [1], VQA-Introspect [2], IV-VQA/CV-VQA [3]and VQA-Rephrasings [4]. I suggest the authors to add discussions about these works, to highlight the key differences and contributions in AdVQA.

[1] Tejas Gokhale, Pratyay Banerjee, Chitta Baral, and Yezhou Yang. Vqa-lol: Visual question answering under the lens of logic. ECCV, 2020.

[2] Ramprasaath R Selvaraju, Purva Tendulkar, Devi Parikh, Eric Horvitz, Marco Ribeiro, Besmira Nushi, and Ece Ka- mar. Squinting at vqa models: Interrogating vqa models with sub-questions. CVPR, 2020.

[3] Vedika Agarwal, Rakshith Shetty, and Mario Fritz. Towards causal vqa: Revealing and reducing spurious correlations by invariant and covariant semantic editing.

[4] M Shah, X Chen, M Rohrbach, and D Parikh. Cycle- consistency for robust visual question answering. In CVPR, 2019

`L34-35, AdVQA probably is not the first work to have human adversarial approach in Multimodal space`

VLEP [5] is a related work in video+language domain, which leverages human-and-model-in-the-loop to collect harder event candidates for future event prediction task. I suggest the author to add discussion about this work.

[5] JieLei, Licheng Yu,Tamara L Berg, and Mohit Bansal. What is more likely to happen next? video-and-language future event prediction. In EMNLP, 2020

`Detailed question type analysis`

In ANLI, the authors have provided detailed analysis on the collected adversarial statements (Table ?? and Section ?? of ANLI) to provide insights on (1) What kind of statements would successfully attack the model and (2) Distribution of different failure cases to understand what human annotators focuses on when attacking the model.

I see that some statistics are provided in Figure 5/6 and Table 5. However, I would be more interested in analysis on detailed question types that is similar to Section ?? in ANLI.

For example, since "Other" questions contribute to more than 40% of AdVQA questions, are these other questions more about testing model's reasoning skills or about recognition of small/novel objects in the image. What kind of reasoning questions are there? Are there more positional reasoning than relational reasoning?

`Answers in AdVQA`

(1) L222-224 suggest that the low performance can be from the answer distribution differences between training and testing data. When further finetune on AdVQA val set, what's the performance on VQA v2?

(2) I also want to understand how much does the answer distribution shift contribute to the low performance.

First, there are questions that are not answerable by VQA v2 answer vocabularies. What's the upper bound of model performance on val and test set if using the VQA v2 answer vocabularies in Pythia?

Second, can we keep adding more training data from AdVQA and see when does the model performance saturate? This may not be easy as no training set is collected for AdVQA. Maybe we can keep a smaller set of test set aside and use all others as training data. I will be happy to discuss more about this point.

`BERT performance on AdVQA`

BERT performance on AdVQA test set seems to be on par with multimodal models. Does this suggest that the multimodal models are not paying attention to the visual signals at all? Or at least it is not paying attention to the image regions that really matters to answer the AdVQA questions. Maybe some visualizations of attention maps could be helpful to understand this result.

`Differences between AdVQA and ANLI`

AdVQA largely follows ANLI, but differs in several data collection settings. I hope to learn from the authors about the reason behind these differences.

(1) Human generated adversarial training data is provided in ANLI, but not in AdVQA. As L222-224 suggested, the training data may be helpful in improving performance on AdVQA.

(2) ANLI was collected for multiple rounds, but AdVQA only collect one round. This point is somewhat related to (1). Without training data, we could not have a stronger model for the second round of attack.

(3) ANLI include diverse context from multiple domains, while AdVQA collects on images from COCO only

`Best Practice with AdVQA`

As the VQA v2 performance is saturating. Would it still make sense to report VQA v2 performance, or should the future work just report results on AdVQA? What about the other stress test dataset?

**Time Spent Reviewing:**

3

---

> ### Author Response · Authors · 2021-08-10
> **Response to Reviewer JzmN**
>
> Thank you for your thoughtful comments, this is great feedback that will help make the paper better!
>
> * We are sorry that we missed citing the VQA stress tests you mention, and we will add them to the final version if accepted. The efforts you mentioned check for robustness along specific axes (e.g., consistency in logic, to question paraphrases, etc.). In AdVQA we have a more holistic and general dataset of questions that push the boundaries of SOTA models in general. In that sense, we take these datasets to be complementary.
> * L34-35: We will rephrase accordingly and cite the papers you mention, thanks for pointing them out!
> * Regarding more detailed analysis of the question types: this is a great suggestion, thanks! Below we provide our insights from our analysis based on your suggestions and the discussions in the paper. We will expand this analysis and add it in the final version if accepted.
>    - Using Rosetta [6] extracted OCR tokens, we find that 15.4% of the val questions are solvable using OCR, suggesting that some of the AdVQA questions require scene text reading and understanding capabilities.
>    - Here is a detailed breakdown of question types based on the categories similar to [3]. We specifically check for the category’s presence at the start of the question:
> ```
> Question Type | full val set (%) | “others” (%)
> what is       | 20.67            | 36.48
> what color    | 4.61             | 9.16
> what kind     | 1.41             | 2.79
> what are      | 2.01             | 3.80
> what type     | 1.44             | 2.87
> is the        | 8.57             | 1.48
> is this       | 0.98             | 2.33
> how many      | 25.28            | 0.27
> are           | 4.09             | 0.82
> does          | 2.15             | 0.16
> where         | 0.84             | 1.66
> is there      | 0                | 0
> why           | 0.04             | 0.01
> which         | 0.59             | 3.38
> do            | 0.51             | 0
> what does     | 2.05             | 4.07
> what time     | 1.23             | 2.41
> who           | 0.59             | 1.16
> what sport    | 0.13             | 0.27
> what animal   | 0.48             | 0.97
> what brand    | 1.41             | 0.28
> ```
>     - We also sampled 50 random examples from the validation set to check what capabilities AdVQA questions require for answering. If more than one category is required for answering the questions, we include the question in all those categories. The categorization is based on the answer (i.e., what answering the question requires), rather than the question. Based on our analysis, we can see that AdVQA does require multiple kinds of reasoning capabilities in a model to solve it:
>         - 7 questions required color understanding.
>         - 12 questions’ answers were common objects (80 common object classes in COCO)
>         - 15 questions required reading or understanding text for answering
>         - 12 questions required positional understanding and reasoning.
>         - 4 questions required understanding uncommon objects
>         - 7 questions required external knowledge in some form.
>         - At least 6 questions were directly solvable by common sense reasoning.
>
> - Answers in AdVQA: as an upper bound using Pythia’s vocabulary on AdVQA, as discussed in L278-282, 77.2% of AdVQA val set’s questions are answerable using Pythia’s vocabulary suggesting that answer distribution difference might not be a major contributing factor to low performance as the current state-of-the-art models’ performance is very far away from 77.2% as even the best model in our Table 3 only achieves 33.33%.
>
> - BERT performance on AdVQA: Thank you for your suggestions. In comparison to multimodal models, BERT is very good at picking up on  natural "biases" in AdVQA, which it probably learns from its pretraining on large scale text corpora. As seen in Table 4, the category-wise performance of BERT compared to multimodal is different as multimodal models perform better on numbers and others category while lagging on yes/no category. BERT’s performance is similar to majority class performance and majority class performance on all categories suggesting that BERT picks up on the dataset biases very well. We confirm this by doing an extensive analysis on BERT and VisualBERT’s predictions on AdVQA and we will include this analysis in the the final version if accepted:
>
>    - 68.4% of the times BERT predicts “no” majority answer compared to 49.6% of the times for VisualBERT when the question is of “yes/no” type.
>    - For numeric questions, 56.3% times BERT predicts majority answer “2” whereas VisualBERT only predicts it 33.3% times.
>    - For questions requiring reading text (keywords “written”, “sign”, “text”), BERT mostly predicts “stop”.
>    - For questions related to brand BERT mostly predicts “nike”.
>    - For countries, BERT mostly answers “usa”.
>    - For color, mostly predicts “black”
>    - For questions asking what the person is holding predicts “tennis racket”.
>    - For cities, BERT mostly predicts “new york”.
>
> - Differences between AdVQA and ANLI:
>    1) The main reason for this was budget: this dataset was already very expensive to create. For similar budgetary reasons, the ANLI training dataset is actually not fully validated (see section 2.1 of the paper), which means that their training data is not necessarily of the best quality and may need to be filtered.
>    2) We agree! We hope to do multiple consecutive rounds in the future. Even without training data, we think the community should be able to train up more robust models that can do well on this dataset, which can then be put in the loop for further data collection.
>    3) The main reasons for sticking to COCO were licensing and accessibility: the community knows where to find COCO images and how to handle them. We appreciate your suggestion of exploring other domains, and we hope to be able to do this in subsequent rounds of data collection.
>
> - Best Practice with AdVQA: This is an excellent question. We would say that people should still report VQAv2 results, for the simple reason that we don’t want models to improve AdVQA performance *at the expense of* VQAv2 performance. We would encourage the community to report results on as many stress tests as possible.

---

### Decision · Program_Chairs · 2021-09-27

**Decision:**

Accept (Poster)

**Comment:**

This paper introduces a new Adversarial VQA dataset collected with human-and-model in the loop by directly asking humans to write questions to attack SOTA VQA models. After author rebuttal, it has received 4 accept recommendations. All the reviewers are happy about the paper, and agree that this is a solid new benchmark worth sharing with the community, and has the potential to be the next generation of a generic VQA dataset for testing future VQA methods. Therefore, the AC is happy to recommend acceptance of the paper.